# The Role of Facebook^®^ in Promoting a Physically Active Lifestyle: A Systematic Review and Meta-Analysis

**DOI:** 10.3390/ijerph19169794

**Published:** 2022-08-09

**Authors:** Federica Duregon, Valentina Bullo, Andrea Di Blasio, Lucia Cugusi, Martina Pizzichemi, Salvatore Sciusco, Gianluca Viscioni, David Cruz-Diaz, Danilo Sales Bocalini, Alessandro Bortoletto, Francesco Favro, Cristine Lima Alberton, Stefano Gobbo, Marco Bergamin

**Affiliations:** 1Department of Medicine, University of Padova, Via Giustiniani, 2, 35128 Padova, Italy; 2Department of Medicine and Sciences of Aging, G. d’Annunzio University of Chieti-Pescara, Via dei Vestini, 31, 66100 Chieti, Italy; 3Department of Biomedical Sciences, University of Sassari, Viale San Pietro 43/B, 07100 Sassari, Italy; 4GymHub S.r.l., Spin-off of the University of Padova, Via O. Galante 67/a, 35129 Padova, Italy; 5Department of Health Sciences, Faculty of Health Sciences, University of Jaén, E-23071 Jaen, Spain; 6Laboratorio de Fisiologia e Bioquimica Experimental, Centro de Educacao Fisica e Deportos, Universida-Federal do Espirito Santo (UFES), Av. Fernando Ferrari, 514, Goiabeiras, Vitoria 29075-910, Brazil; 7Physical Education School, Federal University of Pelotas, Rua Luís de Camões, 625, Pelotas 96055-630, Brazil

**Keywords:** Facebook, social network, physical activity, health benefits, active lifestyle, review, meta-analysis

## Abstract

Background: it is well known in literature that sedentary lifestyle contributes to worsening people’s health. This issue highlights the need for effective interventions to promote an active lifestyle. Research suggested multilevel intervention strategies to promote adherence to recommended physical activity levels, including the use of social networks that may simplify access to health notions. Being Facebook^®^ the most extensive worldwide social network, this document aimed to analyze the current body of evidence on the role of Facebook^®^ in the promotion of physical activity. Methods: eighteen manuscripts were considered eligible for this systematic review, and it was performed a meta-analysis (PRISMA guidelines) for overall physical activity parameters in eleven out of eighteen studies. Results: significant improvements were detected in the total amount of physical activity. In parallel, an increase in other parameters, such as cardiovascular, body composition, and social support, were found. The aerobic training, with supervised and tailored modalities, showed more considerable improvements. Conclusions: this study showed that Facebook^®^ might be considered a feasible and accessible approach to promoting regular exercise practice and achieving health benefits indicators. Future research on the cross-link between physical activity and social network management could also focus on strength training to verify if a more structured intervention would show an effect.

## 1. Introduction

Physical inactivity could be defined as “the 20th-century pandemic” with serious health implications. The most influential aspect is the sedentary lifestyle, which increases the risk of mortality for all causes, doubles the risk of developing cardiovascular disease, diabetes, and obesity, and increases the risk of colon and breast cancer, hypertension, osteoporosis, dyslipidemia, and mood disorders [1]. Moreover, the risk of mortality in people who sat more than 8 h per day or were less active than 2.5 MET-hours per week is similar to the smoker and obese individuals [2]. In 2013, the worldwide economic impact of sedentary was about $67.5 billion through healthcare expenditure and productivity losses [3]. These thoughts highlight the need for effective interventions to promote a physically active lifestyle.

Research reports motivation, encouragement, and self-efficacy [4,5,6], environmental factors [7], and affective judgments in physical activity (PA) participation [8] as determinants of physically active behavior.

Another established interpersonal determinant of physical activity is social support [9]. The social support network size seems to be a contributing factor in adults [8] and older adults. In fact, in this age span, those who perceive having social support have a 41% of increased odds of meeting physical activity guidelines [9]. In line with these considerations, literature reported that socially isolated adults had multiple unhealthy behaviors, including physical inactivity [10].

Effective interventions to promote regular PA can be classified into four general levels of impact: individual, community, communication environment (such as interventions provided through information and communication technologies), physical environments and policy [11]. Even though there is no consensus about the best strategies to change physical activity behavior [12], multilevel intervention strategies [7] appear to be applied to promote adherence to recommended physical activity. In support of this thought, in the World Health Organization (WHO) European policy framework [13], for health and well-being, there were considered different intervention areas to achieve the goals set, including improving access to physical activity and offers.

Social networks are large-scale platforms for information delivery that simplify access to health notions [14]. The possibility of reaching a large audience and the absence of a schedule to enjoy content make social networks cost-effective. The interaction among subscribers enhances efficaciously social support [15], described previously as a determinant of physically active behavior. Moreover, experimental interventions used gamified social networking platforms to improve participation in PA and exercise [16].

To the best of authors’ knowledge, only two reviews [17,18] investigated the effect of social networks in PA promotion; however, they also considered a wide spectrum of health determinants as tobacco and alcohol consumption, dietary intake and sedentary behavior. Moreover, Facebook, Twitter, and health-specific social networks were pooled. Instead, this study focused on only one social network solely to standardize the intervention to be analyzed to a single social model with monitoring and supervision of PA. The authors chose Facebook^®^ as the most extensive worldwide social network service—in the first quarter of 2019, it had 2.38 billion monthly active users; moreover, it allows different ways of interaction, such as the creation of private groups or the launch of conversations between participants, that other social networks do not have. For these reasons, the aim of this study is to understand the role of Facebook in PA and PA-related behaviors promotion, reviewing studies that have investigated healthy parameters and psychosocial variables in individuals from 18 to 65 years.

## 2. Materials and Methods

### 2.1. Study Design

This is a systematic qualitative review and meta-analysis of the literature that aimed to summarize and analyze the influence of Facebook support on the PA practice and how this can improve physical well-being in the general population. The Preferred Reporting Items for Systematic Reviews and Meta-Analyses (PRISMA) guidelines and flow chart diagram were used to report this systematic review and meta-analysis [19,20].

### 2.2. Literature Research

The literature research was performed from January to 21 February 2020. Only articles published after 2004 were taken into consideration. The keyword “social media” OR “social network” OR “Facebook” was associated with “exercise” OR “physical activity” OR “physical exercise” OR “training” OR “aerobic exercise”. The research was carried out on the online database PubMed including MEDLINE, Scopus, and SPORTDiscus. In addition, references were examined in each eligible article, and further relevant manuscripts were screened when a positive match was observed.

### 2.3. Eligibility Criteria

To be included, articles need to meet the following criteria, according to PICOS model [19,20]: (a) including subjects aged between 18 and 65 years; (b) delivering a supervised or a not supervised PA training intervention, with the support of Facebook usage; (c) the presence or not of the control group, with different characteristics (sedentary, other exercise intervention); (d) at least a physical activity quantification; (e) randomized controlled trials (RCTs), non-RCTs, quasi-experimental design (Table 1). All individuals were considered, regardless of race, ethnicity, sex characteristics, and health status. Furthermore, only studies published in English in peer-reviewed journals were considered eligible. Studies that did not evaluate outcomes pre- and post-intervention, cross-sectional studies, reviews, commentaries, perspective studies, editorials, and case reports were excluded. Published abstracts, dissertation materials, or conference presentations were not considered eligible documents.

### 2.4. Study Quality Assessment

The quality of the studies was assessed applying an adapted nine criteria checklist provided by the Cochrane Collaboration Back Review Group [21]. As in previous systematic reviews [BLIND FOR REWIERS], the checklist had to be marginally adapted to rate the strength of the evidence.

Each study in the review was scored based on the following nine criteria: (1) “Was the method of randomization adequate?”; (2) “Were the groups similar at baseline regarding the outcome measures?”; (3) “Were inclusion and exclusion criteria adequately specified?”; (4) “Was the drop-out rata described adequately?”; (5) “Were all randomized participants analyzed in the group to which they were allocated?”; (6) “Was compliance reported for all groups?”; (7) “Was intention-to-treat analysis performed?”; (8) “Was the timing of outcomes assessment similar in all groups?”; (9) “Was a followed-up performed?”.

Whether the paper provided a satisfactory description, a positive value was assigned (+). If the criterion description was considered absent, unclear, or lacked the specified content, a negative value was assigned (−). A study was qualitatively judged as high quality if it showed a positive score on 5 out of 9 criteria; otherwise, it was considered a low-quality study.

### 2.5. Data Extraction and Synthesis

Three researchers (F.D., V.B., S.G.) independently examined all abstracts of the sourced studies. Several studies were analyzed in more detail to be included in the review. During the reference screening of the included items, additional articles were potentially sourced. Independent searches were then combined, compared, and reviewed for applicability, whereas a consensus was made regarding study inclusion. The review process was repeated in case of discrepancies, and a fourth researcher (S.G.) was consulted. Quality assessment using the modified Cochrane methodological quality criteria was then independently applied and discussed before final quality scores were assigned (Table 2). The same researchers who screened titles, abstracts, full texts, and references performed the quality assessment. Several domains were identified for the categorization of the study results. In particular, studies were analyzed in regard to “PA parameters”, “anthropometric parameters”, “cardiovascular parameters”, “diet parameters”, “body composition”, “lipid profile and glucose tolerance”, and “psychosocial parameters”.

### 2.6. Data Analysis

Meta-analysis was performed for the overall PA parameter, using random-effects models with confidence intervals set at 95%; effect size (ES) was calculated through Review Manager 5.4 software (Copenhagen, Denmark, The Nordic Cochrane Centre, The Cochrane Collaboration, 2020). The ES was calculated as standardized mean difference ΔMean/SDpooled, where ΔMean is the difference between the post-intervention mean of the Facebook (FB) Group and Control Group, and SDpooled is the mean of the post-intervention standard deviation, summarizing the different tests evaluating the same parameter. Overall, ES consisted of FB group compared with all the different control groups, including established Control Group (CG), Paper Group (PG), Phone Conference Group (PhC-G), Standard Walking Intervention Group (SWI-G), and Exercise Group (EG). The ES is a measure of the effectiveness of a treatment, and it helps determine whether a statistically significant difference is a difference of practical concern. The interpretation was performed according to guidelines by Cohen [38] and Sawilowsky [39], where an ES value of 0.20 indicates a small effect, ES of 0.50 indicates a medium effect, ES higher than 0.80 indicates a large effect, ES value of 1.20 indicates a very large effect, and an ES value of 2.0 indicates a huge effect.

## 3. Results

A total of 4419 studies resulted from the literature search. Firstly, duplicates were removed, then 3795 records were screened. Based on inclusion and exclusion criteria, 18 articles were considered eligible for this systematic review (Figure 1); one study was excluded due to unavailable data [40].

Quality appraisal classified twelve studies as high quality [18,23,24,25,29,31,32,33,35,36,37,41], while six as low quality [22,27,28,30,34,41]. From a methodological perspective, randomization was applied in twelve studies [18,23,25,29,30,31,32,33,35,36,37,41], and seven [22,23,28,31,32,33,41] showed similarity of the groups’ participants at baseline. All studies adequately reported inclusion and exclusion criteria, except for two [22,34]. Moreover, twelve studies [18,22,23,24,25,26,30,32,33,35,37,41] reported the timing of outcomes assessments and the compliance [18,23,24,25,26,27,29,31,32,33,35,36] of the intervention. All studies reported drop-out ratio, except for Rote et al. and Wang et al., while seven studies [18,24,25,29,31,36,37] applied the intention-to-treat analyses. Finally, four studies [18,24,25,35] performed follow-up evaluations, and only two studies [18,29] performed a single blinding evaluator procedure (Table 2).

Sample sizes ranged from 8 to 375 subjects, with an average of 33.4 years old. In six studies, participants had chronic diseases, in particular, in three studies [23,26,36], there were cancer survivors, in two studies [30,31] individuals with overweight or obesity, and one study [27] include individuals with obesity and depressive disorders. Interventions lasted from 4 to 24 weeks, and the majority of the studies proposed a structured intervention of physical activity, predominantly aerobic training. Only one study [29] guaranteed a supervised physical exercise session (1 h per week) with a trainer. Table 3 summarizes the characteristics of the different protocols. Finally, Table 4 reported all results of the included studies.

### 3.1. Physical Activity Parameters

All included studies analyzed physical activity parameters, adopting mainly pedometers and accelerometers, or administering surveys (Figure 2); only Naslund et al. used a field test. Two investigations showed an increase in the amount of physical activity, within facebook group (FB-G), by 20.6% [29] and 45.5% [37] after 12 weeks of a structured program; also, two investigations presented an amelioration by 89.2% [18] and 39.4% [29] in FB-G compared to CG (ES = 0.22) after 7 and 12 weeks, respectively. Valle et al. reported increased physical activity in all analyzed parameters, particularly in the light physical activity (LPA), where the FB-G-fitnet ameliorated by 197.1%, likened to FB-G-comparison. ES of 0.41 confirmed this result showing a small to medium effect in favor of the FB-G for which there was a more complete intervention. Moreover, an ES of 0.73 indicates a medium to large effect of FB-G in comparison with PG. As a matter of fact, eight weeks of intervention with Facebook support demonstrated an improvement by 7.8% [32] in light activity, by 6.1% [32] in moderate-lifestyle, and by 52.2% [25] in moderate-to-vigorous activity (MVPA). Also, Chee et al., after 16 weeks of a training program, showed a significant increase in the number of steps per day by 84.5% in FB-G compared to PG; this outcome was confirmed after a 2-month follow-up by 58.1% [35]. In 8 weeks of intervention for university students, comparing FB-G to a standard exercise group, it was found an increase of PA by 33.5% [34] and 135.5% [33], respectively, within and between conditions. In addition, Willis et al. showed amelioration in met calorie goal by 44% in favor of the Phone Conference Group (ES = −0.21) [31]. Finally, only two studies reported a worsening within FB-G by 4.2% [24] in the number of steps, by 16.7% [24] in MVPA, and by 58.4% [28] in Leisure Vigorous Activity (VA).

### 3.2. Anthropometric Parameters

Twelve studies analyzed the effect of physical activity, supported by Facebook usage, on anthropometric parameters. In particular, interventions lasted from 12 to 24 weeks, reported a significant reduction in body mass index (BMI) (0.8% [29], 4.6% [30]), weight (0.7% [29]; 4.8% [30]), and waist circumference (WC) (1.3% [29]; 4.7% [30]), compared with CG. Valle et al. reported a significant reduction in weight (7.8%) and BMI (6.3%) within FB-G-fitnet [36], where an administrator encouraged the interaction via social network. Also, Chee et al. found a significant decrease within the group in all the evaluations, with the FB group that achieved a reduction in body weight (7.3%), BMI (7.4%), WC (4.5%), hip circumference (HC) (2.4%), and waist-hip ratio (WHR) (2.3%) [35]. Moreover, in the latter study, there was a significant additional reduction in these evaluations (weight: 4.4%, WC: 2.7%, HC: 1.4%, WHR: 1.1%) after a 2-month follow-up. Contrarily, one study [28] showed a significant reduction in BMI (1.6%) within the control group.

### 3.3. Cardiovascular Parameters

Cardiovascular parameters were investigated in eight out of eighteen studies with different methods. Four studies [23,25,26,41] used the YMCA step test to determine the cardiorespiratory fitness (CRF), three studies [24,30,35] evaluated diastolic and systolic pressure, while one study [29] also measured the resting heart rate and the augmentation index (AI). Only Chee et al. found a significant improvement within-group comparison, with FB-G that decreased both systolic (5.1%) and diastolic (6.2%) pressure; these data were confirmed after 2-month follow-up with a significant additional reduction of 3.1% and 3.8%, respectively [35]. Mixed results were found by Looyestyn et al., where a higher reduction of one-minute post-test (YMCA test) heart rate, not statistically significant, was reported in PG compared with FB-G. On the other hand, the same Paper Group worsened after a 5-month follow-up, while the FB-G confirmed the heart rate reduction [25]. Conversely, another study [26], analyzing the heart rate, showed a decrement in both Facebook Groups after ten weeks of intervention, while FB-G compared with CG in other two studies [29,30] demonstrated an improvement ranged from 0.1% to 11.9% in all cardiovascular evaluations.

### 3.4. Diet Parameters

The influence of physical activity, integrated with a social network, in eating habits was compared in six investigations [23,24,28,29,30,31]. After 12 weeks of intervention, Torquati et al. reported a significant increase in fruit and vegetable intake (26.5%) within the healthy nurses’ Facebook Group; however, this data was not confirmed after a 6-month follow-up [24]. Another two studies found statistically significant changes in diet parameters. In particular, the first [28] that considered the junk food intake, both days per week and times per day, detected a reduction of 39.4% and 33.3%, respectively, within FB-G. The latter [29] showed an increase of 19.2% within FB-G in diet quality score and 28.6% in vegetable servings per day between groups comparison. Finally, the other two studies [30,31], whose sample comprised individuals with overweight or obesity, did not report significant ameliorations in all parameters.

### 3.5. Body Composition

Six [23,26,29,30,35,41] out of eighteen studies examined body composition, using the bioelectrical impedance analysis. A statistically significant amelioration in body fat mass (BFM) was detected in three studies after 12 weeks (2.5% [29]), 16 weeks (17.1% [35]), and 24 weeks (2.6% [30]) of physical activity intervention in FB-G compared with CG [29,30] or PG [35]. Chee et al. also found a significant improvement within FB-G in BFM expressed in kilograms or percentual changes (22.7%). The latter data were confirmed after a 2-month follow-up, with a further BFM reduction of 10.4% in FB-G, both within and between groups comparison [35]. Moreover, a significant increase (1.1%) in lean mass was assessed after 24 weeks of a combined program consisting of physical activity practice and Total Wellbeing Diet for participants with overweight or obesity [30].

### 3.6. Lipid Profile and Glucose Tolerance

Three studies [29,30,35] investigated the impact of physical activity on lipid profile and glucose parameters, such as insulin and fasting glucose. Comparing FB-G with CG, Ashton et al. showed a statistically significant improvement in total cholesterol (10.3%), low-density lipoprotein (LDL) (18.2%), and total cholesterol/high-density lipoprotein (HDL) ratio (9.1%) after 12 weeks of supervised physical activity, combined with a home-based resistance training [29]. Another study [30] compared FB-G with CG but did not find meaningful differences in all parameters, despite the long duration of the intervention. Only triglycerides improved in FB-G (15.4%), contrarily for CG that worsened (8.3%), even though non-significantly. The same study [30] that also considered glucose parameters found a significant decrease in fasting glucose of 7.3% in favor of FB-G compared with PG. Finally, Chee et al., that focused the content of the FB posts on walking activity, detected significant enhancements within FB-G in all evaluations; specifically, in total cholesterol (14.4%), LDL (21.4%), HDL (15.8%), Triglycerides (34.9%), and Fasting glucose (18.1%) [35]. These improvements were confirmed after a 2-month follow-up where FB-G maintained significant changes respect to baseline. Nevertheless, comparison between post-intervention and follow-up evaluations showed significant worsening in above mentioned parameters.

### 3.7. Psychosocial Parameters

Psychosocial parameters were investigated by six studies [23,25,26,32,34,41], using psychometrically validated questionnaires, such as The Self-Efficacy Barriers to Exercise Measure, The Exercise Attitude Questionnaire-18, The Intrinsic Motivation Inventory, The Social Support for Exercise Survey. After eight weeks of intervention, social support significantly improved within FB-G (18.6% [25]) and compared with PG (30.8% [32]); nevertheless, these data were not confirmed after a 5-month follow-up [25]. Conversely, Pope et al. [26] found a significant worsening in social support (10%) in Facebook Group1, in which there was a more detailed physical activity monitoring with smartwatch compared to Facebook Group2. This data countertrend was also confirmed in another study [34], which showed a significant improvement in the competence (20.8%) in Facebook2 Group with 1 h of exercise program, compared with Facebook1 Group that performed 3 h per week of fitness class. In the same study [34], also enjoyment significantly increased (13.7%) in the Facebook2 Group, in within condition. Finally, mixed results were found by Joseph et al.; specifically, it was detected a significant increase in the outcome expectation (6.6%) in PG compared with FB-G, while self-regulation ameliorated considerably in FB-G (89.9%), both within and between groups comparison [32].

**Table 4 ijerph-19-09794-t004:** Results.

Study	Groups (n)	Results
Todorovic J et al. (2019) [22]	FB-G (311) vs. CG (64)	PA parameters ^ IPAQ (met/min/week) ↑: FB-G (+14.4); CG (−22.4)
Pope ZC et al. (2019) [41]	FBS-G (19) vs. FB-G (19)	PA parameters ^ SB (min/day) ↓: FBS-G (+2.9); FB-G (−2.1) LPA (min/day) ↑: FBS-G (−6.9); FB-G (+16.3) MVPA (min/day) ↑: FBS-G (+110.7); FB-G (+44.6) Diet parameters ^ Daily Kcaloric consumption (cals) ↓: FBS-G (−0.9); FB-G (−4.6) Daily Fruit intake (cups) ↑: FBS-G (+36.7); FB-G (−7.0) Daily Vegetable intake ↑: FBS-G (+14.5); FB-G (−0.7) Daily Whole Grain intake (oz. equivalents) ↑: FBS-G (+9.8); FB-G (+93.0) Daily SSB Consumption calories ↓: FBS-G (+47.1); FB-G (−7.5) Psychosocial parameters ^ Self-efficacy (sc) ↑: FBS-G (+33.7); FB-G (+28.7) Social-support (sc) ↑: FBS-G (+36.5); FB-G (+19.0) Enjoyment (sc) ↑: FBS-G (+9.3); FB-G (+5.7) Barriers (sc) ↓: FBS-G (−1.7); FB-G (−4.5) Outcome expectancy (sc) ↑: FBS-G (+8.3); FB-G (+7.7) Intrinsic motivation (sc) ↑: FBS-G (+13.7); FB-G (+12.3) Anthropometric parameters ^ Weight (kg) ↓: FBS-G (−0.2); FB-G (−0.5) Body composition ^ BFM (%) ↓: FBS-G (+11.1); FB-G (+1.7) Cardiovascular parameters ^ Cardiorespiratory fitness (heart rate) ↑: FBS-G (+2.6); FB-G (−2.0)
Torquati L et al. (2018) [24]	FB-G (47), 6-Mo F-Up (27)	PA parameters Steps (n/d) ↑: FB-G (−4.2) *, 6-Mo F-up (−6.2) SB (%) ↓: FB-G (−0.9), 6-Mo F-up (+2.1) SB (min) ↓: FB-G (−4.6), 6-Mo F-up (0.0) LPA (%) ↑: FB-G (+2.1), 6-Mo F-up (−1.5) LPA (min) ↑: FB-G (−2.4), 6-Mo F-up (−5.0) MVPA (%) ↑: FB-G (−16.7) *, 6-Mo F-up (0.0) Anthropometric parameters Weight (kg) ↓: FB-G (−0.1), 6-Mo F-up (−7.6) BMI (kg/m2) ↓: FB-G (−0.4), 6-Mo F-up (−7.4) WC (cm) ↓: FB-G (0.0), 6-Mo F-up (−6.6) Cardiovascular parameters SBP (mmHg) ↓: FB-G (−0.4), 6-Mo F-up (−1.6) DBP (mmHg) ↓: FB-G (−1.5), 6-Mo F-up (−3.5) Diet parameters Energy intake (kJ/d) ↓: FB-G (+2.3), 6-Mo F-up (−8.6) ARF score (diet quality) ↑: FB-G (+0.6), 6-Mo F-up (−1.2) Fruits and vegetables (%) ↑: FB-G (+26.5) *, 6-Mo F-up (−9.7) Discretionary food (% energy) ↓: FB-G (−2.9), 6-Mo F-up (−13.7)
Looyestyn J et al. (2018) [25]	FB-G (41) vs. PG (48); FB-G 5-Mo F-Up (29) vs. PG 5-Mo F-Up (29)	PA parameters MVPA (min/week) ↑: FB-G (+52.2) *,**; PG (+25.4) * FB-G 5-Mo F-Up (−2.7); PG 5-Mo F-Up (−31.3) Psychosocial parameters SE of BEM (sc) ↑: FB-G (−7.3); PG (+4.0) FB-G 5-Mo F-Up (+7.2); PG 5-Mo F-Up (−4.7) Exercise attitude (sc) ↑: FB-G (+2.2); PG (+3.0) FB-G 5-Mo F-Up (−1.7); PG 5-Mo F-Up (0.3) Social support and exercise survey (sc) ↑: FB-G (+18.6) *; PG (+2.8) * FB-G 5-Mo F-Up (−12.4); PG 5-Mo F-Up (−4.6) Cardiovascular parameters HR (bpm) ↓: FB-G (−4.5); PG (−6.6) FB-G 5-Mo F-Up (−0.8); PG 5-Mo F-Up (+3.9)
Pope ZC et al. (2018) [26]	FB-G (8)	PA parameters ^ Steps (n/d) ↑: FB-G (+33.6) SB (min/day) ↓: FB-G (−22.8) LPA (min/day) ↑: FB-G (−8.6) MVPA (min/day) ↑: FB-G (+9.7) EE ↑: FB-G (+20.6) Psychosocial parameters ^ Self-efficacy (sc) ↑: FB-G (+3.3) Social-support (sc) ↑: FB-G (+19.9) Enjoyment (sc) ↑: FB-G (+4.7) Barriers (sc) ↓: FB-G (−1.0) Outcome expectancy (sc) ↑: FB-G (−0.3) Anthropometric parameters ^ Weight (kg) ↓: FB-G (−2.9) Body composition ^ BFM (%) ↓: FB-G (−6.2) Cardiovascular parameters ^ Cardiorespiratory fitness (heart rate) ↑: FB-G (0.0)
Pope ZC et al. (2018) [26]	FBS-G (12) vs. FB-G (8)	PA parameters Steps (n/d) ↑: FBS-G (+7.1); FB-G (+7.6) SB (min/day) ↓: FBS-G (+0.5); FB-G (+0.1) LPA (min/day) ↑: FBS-G (+8.2); FB-G (+8.1) MVPA (min/day) ↑: FBS-G (+11.4); FB-G (+25.2) EE ↑: FBS-G (+8); FB-G (+5.7) Psychosocial parameters Self-efficacy (sc) ↑: FBS-G (−10.2); FB-G (−8.2) Social-support (sc) ↑: FBS-G (−10) **; FB-G (+25) Enjoyment (sc) ↑: FBS-G (−3); FB-G (+3.1) Barriers (sc) ↓: FBS-G (0.0) **; FB-G (−14.3) Outcome expectancy (sc) ↑: FBS-G (−4.9); FB-G (0.0) Anthropometric parameters Weight (kg) ↓: FBS-G (+0.4); FB-G (0.0) Body composition BFM (%) ↓: FBS-G (+1.0); FB-G (−2.8) Cardiovascular parameters Cardiorespiratory fitness (heart rate) ↑: FBS-G (−4.3); FB-G (−4.3)
Naslund JA et al. (2018) [27]	FB-G (19)	PA parameters Improved Fitness (>50 m on 6-MWT) ↑: Yes: 4 No: 14 Anthropometric parameters Weight loss (>5%) ↓: Yes: 7 No: 12
Krishnamohan S et al. (2017) [28]	FB-G (22) vs. CG (23)	PA parameters SB (min/day) ↓: FB-G (+7.4); CG (−6.5) Academic MA (min/week) ↑: FB-G (+32.3); CG (−100) Leisure MA (min/week) ↑: FB-G (+0.9); CG (+35.5) Walking/cycling (min/week) ↑: FB-G (+66.5); CG (+36.1) Academic VA (min/week) ↑: FB-G (0.0); CG (0.0) Leisure VA (min/week) ↑: FB-G (−58.4) *; CG (−47.2) Anthropometric parameters BMI (kg/m2) ↓: FB-G (+0.3); CG (−1.6) * Diet parameters Intake of fruit (d/w) ↑: FB-G (−2.3); CG (+33.3) Intake of fruit (s/d) ↑: FB-G (+28.3); CG (+11.1) Intake of vegetables (d/w) ↑: FB-G (+8.5); CG (−10.5) Intake of vegetables (s/d) ↑: FB-G (0.0); CG (+9.1) Outside meals/week ↓: FB-G (−30.9); CG (+10.5) Intake of junk food (d/w) ↓: FB-G (−39.4) *; CG (−6.9) Intake of junk food (t/d) ↓: FB-G (−33.3) *; CG (0.0)
Ashton LM et al. (2017) [29]	FB-G (23) vs. CG (23)	PA parameters Steps (n/d) ↑ #: FB-G (+20.6) *; CG (+9.3) MVPA (min/week) ↑: FB-G (+39.4) **; CG (+24) Anthropometric parameters Weight (kg) ↓: FB-G (−0.7) **; CG (+1.2) BMI (kg/m2) ↓: FB-G (−0.8) **; CG (+1.2) WC (cm) ↓: FB-G (−1.3) **; CG (+2.2) Body composition BFM (kg) ↓: FB-G (−2.5) **; CG (+5.3) SMM (kg) ↑: FB-G (0); CG (+0.3) Cardiovascular parameters SBP (mmHg) ↓: FB-G (−2); CG (−2.1) DBP (mmHg) ↓: FB-G (−2.3); CG (−1.4) HRrest (bpm) ↓: FB-G (−0.1); CG (−2.8) AI (%) ↓: FB-G (−11.9); CG (−2) Diet parameters Diet quality (ARF total score) ↑: FB-G (+19.2) *; CG (+8.2) Fruit (serves/day) ↑: FB-G (+33.3); CG (+23.1) Vegetables (serves/day) ↑: FB-G (+28.6) **; CG (−2.9) Energy intake (kJ/day) ↓: FB-G (−4.3); CG (−0.7) Proportion of energy from ED-NP foods (%) ↓: FB-G (−24.4) **; CG (−6.2) Proportion of energy from alcohol (%) ↓: FB-G (+11.8); CG (−19.2) Lipid profile and Glucose tolerance Chol-tot (mmol/L) ↓: FB-G (−10.3) **; CG (0) LDL (mmol/L) ↓: FB-G (−18.2) **; CG (+5) HDL (mmol/L) ↑: FB-G (0); CG (0) Chol-tot/HDL-C ratio↓: FB-G (−9.1) **; CG (0) TG (mmol/L) ↓: FB-G (0); CG (0)
Jane M et al. (2017) [30]	FB-G (19) vs. CG (17)	PA parameters Steps (n/d) ↑: FB-G (+28.5) Energy expenditure (kJ/die) ↑: FB-G (−1.7); CG (+1.5) Anthropometrics parameters Weight (kg) ↓: FB-G (−4.8) **; CG (−1.6) BMI (kg/m2) ↓: FB-G (−4.6) **; CG (−1.5) WC (cm) ↓: FB-G (−4.7) **; CG (−1.8) HC (cm) ↓: FB-G (−2.9); CG (−1.3) Body composition BFM (%) ↓: FB-G (−2.6) **; CG (−0.6) Lean Mass (%) ↑: FB-G (+1.1) **; CG (+0.2) Cardiovascular parameters SBP (mmHg) ↓: FB-G (−2.3); CG (+2.8) DBP (mmHg) ↓: FB-G (−0.7); CG (+1.6) Lipid profile and Glucose tolerance Chol-tot (mmol/L) ↓: FB-G (−3.4); CG (+1.8) LDL (mmol/L) ↓: FB-G (−5.3); CG (0) HDL (mmol/L) ↑: FB-G (0); CG (+6.7) TG (mmol/L) ↓: FB-G (−15.4); CG (+8.3) Fasting glucose (mmol/L) ↓: FB-G (−7.3); CG (+6.9) Insulin (mU/L) ↓: FB-G (−1); CG (+1.2) Diet parameters Energy Intake (kJ/day) ↓: FB-G (−18.3); CG (−13.7) Carbohydrate (%): FB-G (−3); CG (+0.1) Protein (%): FB-G (+4.8); CG (+1.3) Fat (%): FB-G (−2); CG (−0.9) Alcohol (%) ↓: FB-G (−0.5); CG (−0.6) Fibre (g): FB-G (−1.7); CG (−1.8)
	FB-G (19) vs. PG (18)	PA parameters Steps (n/d) ↑: FB-G (+28.5); PG (+10.7) Energy expenditure (kJ/die) ↑: FB-G (−1.7); PG (−9.8) Anthropometrics parameters Weight (kg) ↓: FB-G (−4.8); PG (−4.2) BMI (kg/m2) ↓: FB-G (−4.6); PG (−4) WC (cm) ↓: FB-G (−4.7); PG (−3.1) HC (cm) ↓: FB-G (−2.9); PG (−2.8) Body composition BFM (%) ↓: FB-G (−2.6); PG (−1.4) Lean Mass (%) ↑: FB-G (+1.1); PG (+0.6) Cardiovascular parameters SBP (mmHg) ↓: FB-G (−2.3); PG (−0.2) DBP (mmHg) ↓: FB-G (−0.7); PG (−0.1) Lipid profile and Glucose tolerance Chol-tot (mmol/L) ↓: FB-G (−3.4); PG (−1.7) LDL (mmol/L) ↓: FB-G (−5.3); PG (−2.7) HDL (mmol/L) ↑: FB-G (0); PG (0) TG (mmol/L) ↓: FB-G (−15.4); PG (+36.4) Fasting glucose (mmol/L) ↓: FB-G (−7.3) **; PG (−6.5) Insulin (mU/L) ↓: FB-G (−1); PG (+11.4) Diet parameters Energy Intake (kJ/day) ↓: FB-G (−18.3); PG (−13) Carbohydrate (%): FB-G (−3); PG (−3.2) Protein (%): FB-G (+4.8); PG (+3.2) Fat (%): FB-G (−2); PG (0) Alcohol (%) ↓: FB-G (−0.5); PG (−0.3) Fibre (g): FB-G (−1.7); PG (+2.4)
	PG (18) vs. CG (17)	PA parameters Steps (n/d) ↑: PG (+10.7); Energy expenditure (kJ/die) ↑: PG (−9.8); CG (+1.5) Anthropometrics parameters Weight (kg) ↓: PG (−4.2) **; CG (−1.6) BMI (kg/m2) ↓: PG (−4); CG (−1.5) WC (cm) ↓: PG (−3.1); CG (−1.8) HC (cm) ↓: PG (−2.8); CG (−1.3) Body composition BFM (%) ↓: PG (−1.4); CG (−0.6) Lean Mass (%) ↑: PG (+0.6); CG (+0.2) Cardiovascular parameters SBP (mmHg) ↓: PG (−0.2); CG (+2.8) DBP (mmHg) ↓: PG (−0.1); CG (+1.6) Lipid profile and Glucose tolerance Chol-tot (mmol/L) ↓: PG (−1.7); CG (+1.8) LDL (mmol/L) ↓: PG (−2.7); CG (0) HDL (mmol/L) ↑: PG (0); CG (+6.7) TG (mmol/L) ↓: PG (+36.4); CG (+8.3) Fasting glucose (mmol/L) ↓: PG (−6.5) **; CG (+6.9) Insulin (mU/L) ↓: PG (+11.4); CG (+1.2) Diet parameters Energy Intake (kJ/day) ↓: PG (−13); CG (−13.7) Carbohydrate (%): PG (−3.2); CG (+0.1) Protein (%): PG (+3.2); CG (+1.3) Fat (%): PG (0); CG (−0.9) Alcohol (%) ↓: PG (−0.3); CG (−0.6) Fibre (g): PG (+2.4) **; CG (−1.8)
Willis EA et al. (2017) [31]	FB-G (34) vs. PhC-G (36)	PA parameters Steps per day (num)(a) ↑: FB-G (7050.6 ± 3332.1); PhC-G (6810.7 ± 2852.4) PA (min)(a) ↑: FB-G (103.1 ± 149.7); PhC-G (118.8 ± 91.1) Accelerometer (counts/min) ↑: FB-G (+7.6); PhC-G (+6.9) Met calorie goal (%) ↑: FB-G (22 ± 21.2); PhC-G (44 ± 26.2) ** Complete weekly reports (%) ↑: FB-G (34.7 ± 28.9); PhC-G (37.6 ± 27.4) Anthropometric parameters Weight (kg) ↓: FB-G (−5.8); PhC-G (−6.3) BMI (kg/m2) ↓: FB-G (−6); PhC-G (−5.9) WC (cm) ↓: FB-G (−9.7); PhC-G (−3.5) Categories of weight change (%) Gained: FB-G (11.8); PhC-G (11.1) Lost 0 to 4.9%: FB-G (38.2); PhC-G (30.6) Lost 5 to 9.9%: FB-G (17.7); PhC-G (33.3) Lost >10%: FB-G (14.7); PhC-G (13.9) Diet parameters Kcal/die ↓: FB-G (−24.9); PhC-G (−22.9) Carbohydrate (g): FB-G (−16.1); PhC-G (−23.1) Protein (g): FB-G (−15.8); PhC-G (−10.8) Fat (g): FB-G (−36.2); PhC-G (−22.5)
Joseph RP et al. (2015) [32]	FB-G (14) vs. PG (15)	PA parameters Weekly steps (num) ↑: FB-G (−16); PG (−18.9) Sedentary (0–99 ctm) ↓: FB-G (−0.9) **; PG (+1.5) Light activity (100–759 ctm) ↑: FB-G (+7.8) **; PG (+5) Moderate-lifestyle (760–1951 ctm) ↑: FB-G (+6.1) **; PG (−7.3) Moderate activity (1952–5725 ctm) ↑: FB-G (−7.4); PG (+2.3) Vigorous activity (>5725 ctm) ↑: FB-G (+500); PG (−50) Total moderate-to-vigorous activity (>1951 ctm) ↑: FB-G (−5.7); PG (−0.7) Moderate-to-vigorous activity in 10 min bouts or greater (>1951 ctm) ↑: FB-G (+18.2); PG (+9.1) Exercise vital sign (minutes/week) ↑: FB-G (+68.1) *,**; PG (+6.9) Psychosocial parameters Outcome expectations (sc) ↑: FB-G (+3.5); PG (+6.6) ** Self-regulation (sc) ↑: FB-G (+89.9) *,**; PG (+55.1) * Self-efficacy (sc) ↑: FB-G (+4.2); PG (+6.5) Social support from friends (sc) ↑: FB-G (+28.1); PG (+11.4) Social support from family (sc) ↑: FB-G (+30.8) *,**; PG (+14.6) Anthropometric parameters BMI (kg/m2) ↓: FB-G (−0.2); PG (+1)
Rote AE et al. (2015) [33]	FB-G (27) vs. SWI-G (26)	PA parameters Mean weekly steps per day (n) ↑: FB-G (+135.5) *,**; SWI-G (+81.2) *
Maher C et al. (2015) [18]	FB-G (51) vs. CG (59); FB-G 20-week F-Up (44) vs. CG 20-week F-Up (52)	PA parameters PA time (min/week) ↑: FB-G (+89.2) **; CG (+40.6) FB-G 20-week F-Up (−28.8); CG 20-week F-Up (−14.3) Walking time (min/week) ↑: FB-G (+161.4) **; CG (+45.5) FB-G 20-week F-Up (−50.3); CG 20-week F-Up (−16.9) Moderate PA (min/week) ↑: FB-G (+46); CG (+47.8) FB-G 20-week F-Up (−47.9); CG 20-week F-Up (−36.8) Vigorous PA (min/week) ↑: FB-G (+50); CG (+31.7) FB-G 20-week F-Up (+14.1); CG 20-week F-Up (−1.2)
Wang CKJ et al. (2015) [34]	FB3h-G (14) vs. FB1h-G (24)	PA parameters Short form IPAQ (met/min/week) ↑: FB3h-G (+33.5) *; FB1h-G (+4.2) Psychosocial parameters Autonomy (sc) ↑: FB3h-G (−4.9); FB1h-G (+4.8) Competence (sc) ↑: FB3h-G (+12) *; FB1h-G (+20.8) ** Relatedness (sc) ↑: FB3h-G (+8); FB1h-G (+4) Enjoyment (sc) ↑: FB3h-G (0); FB1h-G (+13.7) * Vitality (sc) ↑: FB3h-G (+9.9); FB1h-G (+8.9)
	FB3h-G (14) vs. EG (17)	PA parameters Short form IPAQ (met/min/week) ↑: FB3h-G (+33.5) *; EG (+20.5) * Psychosocial parameters Autonomy (sc) ↑: FB3h-G (−4.9); EG (−3.5) Competence (sc) ↑: FB3h-G (+12) *.**; EG (+10.7) * Relatedness (sc) ↑: FB3h-G (+8); EG (−2) Enjoyment (sc) ↑: FB3h-G (0); EG (0) Vitality (sc) ↑: FB3h-G (+9.9); EG (+9.1)
	FB3h-G (14) vs. CG (7)	PA parameters Short form IPAQ (met/min/week) ↑: FB3h-G (+33.5) *; CG (+18.5) Psychosocial parameters Autonomy (sc) ↑: FB3h-G (−4.9); CG (+3.2) Competence (sc) ↑: FB3h-G (+12) *; CG (+5.5) Relatedness (sc) ↑: FB3h-G (+8); CG (−4.8) Enjoyment (sc) ↑: FB3h-G (0); CG (+14.1) Vitality (sc) ↑: FB3h-G (+9.9); CG (+9.1)
	FB1h-G (24) vs. EG (17)	PA parameters Short form IPAQ (met/min/week) ↑: FB1h-G (+4.2); EG (+20.5) * Psychosocial parameters Autonomy (sc) ↑: FB1h-G (−4.8); EG (−3.5) Competence (sc) ↑: FB1h-G (+20.8) **; EG (+10.7) * Relatedness (sc) ↑: FB1h-G (+4); EG (−2) Enjoyment (sc) ↑: FB1h-G (+13.7) *; EG (0) Vitality (sc) ↑: FB1h-G (+8.9); EG (+9.1)
	FB1h-G (24) vs. CG (7)	PA parameters Short form IPAQ (met/min/week) ↑: FB1h-G (+4.2); CG (+18.5) Psychosocial parameters Autonomy (sc) ↑: FB1h-G (−4.8); CG (+3.2) Competence (sc) ↑: FB1h-G (+20.8); CG (+5.5) Relatedness (sc) ↑: FB1h-G (+4); CG (−4.8) Enjoyment (sc) ↑: FB1h-G (+13.7) *,**; CG (+14.1) Vitality (sc) ↑: FB1h-G (+8.9); CG (+9.1)
	EG (17) vs. CG (7)	PA parameters Short form IPAQ (met/min/week) ↑: EG (+20.5) *; CG (+18.5) Psychosocial parameters Autonomy (sc) ↑: EG (−3.5); CG (+3.2) Competence (sc) ↑: EG (+10.7) *; CG (+5.5) Relatedness (sc) ↑: EG (−2); CG (−4.8) Enjoyment (sc) ↑: EG (0); CG (+14.1) Vitality (sc) ↑: EG (+9.1); CG (+9.1)
Chee HP et al. (2014) [35]	FB-G (35) vs. PG (85); FB-G 2-month F-Up (35) vs. PG 2-month F-Up (85)	PA parameters Step per day (num) # ↑: FB-G (+84.5) *,**; PG (+13.2) * FB-G 2-month F-Up (+58.1) *,**; PG 2-month F-Up (+9.6) * Anthropometric parameters Weight (kg) # ↓: FB-G (−7.3) *; PG (−1.1) * FB-G 2-month F-Up (−4.4) *; PG 2-month F-Up (−0.5) * BMI (kg/m2) # ↓: FB-G (−7.4) *; PG (−1) * FB-G 2-month F-Up (−4.4); PG 2-month F-Up (−0.5) WC (cm) # ↓: FB-G (−4.5) *; PG (−0.7) * FB-G 2-month F-Up (−2.7) *; PG 2-month F-Up (−0.3) * HC (cm) # ↓: FB-G (−2.4) *; PG (−0.4) * FB-G 2-month F-Up (−1.4) *; PG 2-month F-Up (−0.2) * WHR (sc) # ↓: FB-G (−2.3) *; PG (−1.1) * FB-G 2-month F-Up (−1.1) *; PG 2-month F-Up (0.0) * Body composition BFM (kg) # ↓: FB-G (−22.7) *; PG (−3.9) * FB-G 2-month F-Up (−14.1); PG 2-month F-Up (−2.7) BFM (%) # ↓: FB-G (−17.1) *,**; PG (−3) * FB-G 2-month F-Up (−10.4) *,**; PG 2-month F-Up (−2.2) * Lipid profile and Glucose tolerance Chol.tot (mmol/L) # ↓: FB-G (−14.4) *; PG (−3.6) * FB-G 2-month F-Up (−9.6) *; PG 2-month F-Up (−1.7) * LDL (mmol/L) # ↓: FB-G (−21.4) *; PG (−4.9) * FB-G 2-month F-Up (−14.5) *; PG 2-month F-Up (−1.8) * HDL (mmol/L) # ↑: FB-G (+15.8) *; PG (+3.6) * FB-G 2-month F-Up (9.2) *; PG 2-month F-Up (1.8) * Triglycerides (mmol/L) # ↓: FB-G (−34.9) *; PG (−7) * FB-G 2-month F-Up (−19.7) *; PG 2-month F-Up (−4.8) * Fasting glucose (mmol/L) # ↓: FB-G (−18.1) *; PG (−3.2) * FB-G 2-month F-Up (−11.7) *; PG 2-month F-Up (−0.8) * Cardiovascular parameters SBP (mmHg) # ↓: FB-G (−5.1) *; PG (−0.9) * FB-G 2-month F-Up (−3.1) *; PG 2-month F-Up (−0.5) * DBP (mmHg) # ↓: FB-G (−6.2) *; PG (−1.2) * FB-G 2-month F-Up (−3.8) *; PG 2-month F-Up (−0.6) *
Valle CG et al. (2013) [36]	FB-G.fit (32) vs. FB-G.com (34)	PA parameters MVPA (min/week) ↑: FB-G.fit(+51.1) *; FB-G.com(+38.9) * LPA (min/week) ↑: FB-G.fit(+197.1) *,**; FB-G.com(+25.3) Total PA (min/week) ↑: FB-G.fit(+112) *; FB-G.com(+33.4) * Anthropometric parameters Weight (kg) ↓: FB-G.fit(−7.8) *; FB-G.com(−1.9) BMI (kg/m2) ↓: FB-G.fit(−6.3) *; FB-G.com(−1.4)
Cavallo N et al. (2012) [37]	FB-G (67) vs. CG (67)	PA parameters PA total (kcal) ↑: FB-G (+45.5) *; CG (+31.8) PA light (kcal) ↑: FB-G (+5.8); CG (+138.3) PA moderate (kcal) ↑: FB-G (+213.2) *; CG (+105.9) PA heavy (kcal) ↑: FB-G (+96.5) *; CG (+142.5)

Results are shown as (Δ%), except for (a) that are represented as number. Abbreviations: *p* < 0.05 * within group comparison, ** between groups comparison; #: significant difference at baseline (both for post intervention and follow up); ^: no inferential statistics; ↑ or ↓ indicate the desirable direction; (n) indicates the analyzed sample. FB-G: Facebook group; CG: control group; PG: paper group; FB+S-G: Facebook + smartwatch group; PhC-G: phone conference group; EG: exercise group; TM-G: text message group; SWI-G: Standard Walking Intervention Group; PA: physical activity; BMI: Body Mass Index; WC: waist circumference; SBP: systolic blood pressure; DBP: diastolic blood pressure; ARF: Australian Recommended Food; MVPA: Moderate to vigorous physical activity; LPA: light physical activity; MA: moderate activity; VA: vigorous activity; HR: heart rate; SB: sedentary behavior; EE: energy expenditure; 6-MWT: 6-min walking test; BFM: body fat mass; SMM: skeletal muscle mass; AI: Augmentation index; Chol-tot: cholesterol total; LDL; low density lipoprotein; HDL: high density lipoprotein; TG: triglycerides; HC: hip circumference; IPAQ: International physical activity questionnaire; WHR: waist-hip ratio; SSB: sugar-sweetened beverages; n/d: number of steps per day; F-Up: follow up; sc: score; SE of BEM: Self Efficacy of Barriers to Exercise Measure; d/w: days/week; s/d: servings/day; t/d: times/day; ED-NP foods: Energy-Dense, Nutrient poor foods; ctm: count per minutes.

## 4. Discussion

This systematic review focused on the effectiveness of intervention delivered via Facebook in improving physical activity behavior and health outcomes. Particularly, we investigated the role of Facebook usage in PA promotion on healthy parameters and psychosocial variables. The results showed that the Facebook interaction could improve the amount of physical activity, social support, self-regulation, competence and enjoyment, anthropometric, lipid, and glucose parameters compared to a control group or a paper group. Conversely, the data did not confirm this effect when they compared FB-G to another group or with a different exercise treatment. Our results must be read in light of the primary studies’ quality, which showed that most of the studies are of high quality, therefore it can suppose that they used a methodologically correct approach, with a randomization procedure and specific criteria. Conversely, among the low-quality studies, there are pilot studies that need to be implemented in the future to confirm what has already been reported.

In this research analysis, 18 investigations were included; different study designs and different samples, such as healthy adults, cancer survivors, obese or overweight people, underlined the heterogeneity of the studies. All studies reported at least one PA evaluation, and participants involved only used the Facebook social network. Ten studies [23,25,26,30,31,32,33,34,35,36] compared FB-G to other types of physical activity management, eight studies [18,22,27,28,29,30,34,37] to CG with no exercise intervention, while two studies [23,24] had not the comparison group.

In addition, we investigated the efficacy of Facebook exercise interventions in improving the total amount of physical activity with a meta-analytic approach; eleven [18,22,26,29,30,31,33,34,35,36,37] out of eighteen studies were included in the meta-analysis.

### 4.1. Physical Activity Parameters

Sedentary habit is a detrimental phenomenon with a major health effect worldwide [42]. In contrast, it is well known that regular moderate-intensity physical activity seems to attenuate the increased risk of death associated with high sitting time [2]. Nowadays, many different interventions address this issue, including social network involvement that can engage a large number of users at a low cost [43]. Our results revealed that Facebook usage could potentially involve people to be more physically active, not only in comparison to a sedentary group (CG) but also to other groups with less interactive physical activity management. Interventions consisted of promoting, via Facebook, strategies to increase the number of steps per day [35], or to maintain a physically active lifestyle [32], even with team challenge [18] or photo posts and opinion polls [25]. ES analysis confirmed this trend, particularly likening FB-G with PG, for which there were only written form physical activity instructions to follow individually. We can speculate that 8 to 12 weeks of structured program could be sufficient to gain positive effects, even more with an extended period such as Chee et al. that maintained improvements after 2-month follow up [35]. Nevertheless, according to ES analysis, EG intervention seemed to be more effective than FB-G. A physical fitness class with the presence and the supervision of a trainer [34], following a standard intervention [33], could be more effective than web-based involvement [44].

### 4.2. Anthropometric Parameters

Anthropometric parameters were analyzed in many included studies, comparing FB-G with CG, PG, PhC-G, or other FB-G. Nevertheless, few investigations [29,30,35,36] found statistically significant improvements. The twelve studies that carried out anthropometric evaluations had 12.5 weeks of average duration of protocol; hence it could be more advisable for a longer period to obtain a considerable change in weight [45] or waist and hip circumferences. Two studies [27,31] managed 24 weeks of intervention; however, they did not find significant changes in the mentioned parameters. This is probably because they had a sample composed of individuals with obesity, and the program consisted of only one required weight management session or health lesson per week. Also, Jane et al. observed the 24 weeks of intervention and included participants with overweight and obesity, and it was shown significant improvements in weight, BMI, and WC in favor of FB-G compared with CG [30]. In this specific case, participants were instructed to follow the “Total Wellbeing Diet” and issued a pedometer. Therefore, we can speculate that a web-based interaction could not be efficient without a multilevel approach [12], especially with people with obesity. In addition, the specificity of a supervised and tailored program should never be overlooked, not even with social network usages, such as Ashton et al. that found significant changes in all anthropometric parameters proposing a supervised PA and an individualized session with targeted goals [29].

### 4.3. Cardiovascular Parameters

There is consensus that physical activity protects against the development of cardiovascular diseases, and several meta-analyses have concluded that physical exercise has a positive effect on blood pressure levels in both normotensive and hypertensive cases [46]. In this review, eight studies [23,24,25,26,29,30,35,41] analyzed blood pressure and CRF values in an average 13 weeks of intervention, but only Chee et al. showed significant improvements in systolic and diastolic blood pressure within FB-G. Probably, this result could be due to the major attention on the content of the FB posts, focused on the walking activity, the amount of recommended steps per day, and the benefits of walking. It is well known that the beneficial outcomes of aerobic training include improvements in CRF, endothelial function, oxidative stress, and myocardial function [47]. Future research could investigate the longer-term effects of a physical activity intervention on cardiovascular parameters, managed with a social network.

### 4.4. Diet Parameters

Lifestyle behaviors, including diet, physical activity, and sedentary behavior, play an important role in maintaining well-being, and improving these behaviors is considered essential also to reduce the health burden of non-communicable diseases [48]. A recent study [49] conducted a profile analysis of attitudes and barriers that influence these habits to tailor an individual-level intervention that implies attention to diet and PA. In this review, six included studies matched physical activity with diet intervention, and three out of six found significant improvements in diet parameters. Particularly, they used a “dinner disc” to guide main meal portion size [29], and the content of education messages, published in the FB group, was decided by a team of a nutritionist, physical education trainer, and public health specialist [28]. As mentioned above, the multidisciplinary and the specificity of intervention could be more appropriate to change these habits. However, after 24 weeks of participation, individuals with overweight and obesity [30,31] did not obtain improvements, probably due to the different situation compared to a sample of healthy adults and the lack of more incisive appropriate interventions.

### 4.5. Body Composition

Six out of eighteen included studies evaluated body composition, and three of them [29,30,35] showed significant results, particularly in body fat mass. Simultaneously, they significantly improved the total amount of physical activity in favor of FB-G, proposing a prevalent aerobic training supported by social networking. A recent review [50] underlined that exercise has moderate-to-large effects on body composition and physical functioning. In addition, the duration of interventions from 12 to 24 weeks seems to be in line with the literature. Kelley et al. demonstrated that the practice of aerobic training in older adults could significantly modify BFM after 4–5 months [51]. The same authors suggested that resistance training increases muscle mass already after 2–3 months. It is also well known that strength training is a sound method to improve body composition [52]. The articles included in this review did not introduce a structured resistance training intervention, perhaps due to the major management effort via social networks. However, future research could focus on strength training supported by Facebook usage, test the feasibility of this type of intervention, and obtain an overview of the exercise methods finalized on body composition parameters.

### 4.6. Lipid Profile and Glucose Tolerance

In this article, limited findings were reported for lipid and glucose profiles. Indeed, only three studies [29,30,35] focused on these parameters, comparing FB-G with CG or PG. Ashton et al. found significant results in all evaluations after 12 weeks of a structured intervention. We can speculate that supervised aerobic exercise training can increase fitness and improvements in some metabolic outcomes, as well as Finucane et al. explained [53], even though Jane et al. showed a statistically significant result only for fasting glucose outcome, compared with PG after 24 weeks of intervention [30]. Furthermore, there was a positive trend also in other parameters, but not significant enough. Finally, in the mentioned study, participants were obese or overweight, and it is well known that obesity is strongly associated with the development of insulin resistance and, consequently, with metabolic syndrome components [54]. Perhaps, in this case, it could be more efficient a constant supervision, both for PA and diet, not only with social network interaction. Finally, Chee et al. found an improvement in all lipid and glucose parameters within FB-G [35], focusing on walking activity as suggested in another study [55].

### 4.7. Psychosocial Parameters

Psychosocial parameters were largely analyzed through psychometrically questionnaires, investigating social support, self-efficacy, self-regulation, enjoyment, competence, outcome expectations, and barriers. Particularly, social support showed significant results. As a matter of fact, both Looyestyn et al. and Joseph et al. improved this outcome in favor of FB-G, encouraging social interaction and facing discussion topics, such as the development of social support networks to promote PA [25,32]. Conversely, another study [26] found an amelioration on social support in favor of the Facebook Group2, in which participants not used smartwatch as support to PA. We can suppose that intervention less focused on PA (no smartwatch for exercise monitoring) may have influenced the result, making social support more perceived.

Moreover, Joseph et al. showed a significant improvement in self-regulation, for which FB-G ameliorated by 89.2% [32]. We can speculate that proposing an intervention with an extensive range of FB discussion topics, such as developing a PA plan, strategies for maintaining an active lifestyle, and overcoming barriers to physical exercise, could be effective for the self-regulation for PA. Finally, Wang et al., comparing FB-G with CG, EG, and another FB-G, found an interesting result in the competence [34]. In fact, it seems that the FB-G group perceived more competence than the other groups, probably due to the physical fitness class taught by a university lecturer, supported by social networking itself, compared to an ordinary fitness class.

### 4.8. Limitations

In this study, some limitations should be considered. Firstly, in this review, some investigations were pilot studies and, due to their nature, they did not execute inferential statistics, so we cannot compare all evaluations of each other. Secondly, the meta-analysis was performed only on PA parameters because the other outcome evaluations were considered only in a small part of the studies. Thirdly, included study presented high heterogeneity of the methologies to evaluate pA (e.g., pedometers, smartwatch, questionnaire). Finally, the study analysis focused solely on Facebook, as mentioned above, by methodological choice. Therefore, it could be interesting to consider a larger spectrum of parameters that can influence future research outcomes.

## 5. Conclusions

The Facebook networking can be considered feasible and easily accessible to promote the practice of exercise and to increase the total amount of physical activity. Furthermore, considering Facebook^®^ as a large social media platform, one of the biggest worldwide and used in different range age, there could be a global influence on awareness of being physically active. Various strategies, for instance counseling with tips, social networking, challenges, can be employ as good practice to obtain an amelioration on healthy parameters, such as cardiovascular and body composition, and social support, with a great impact on public health. However, it is important to maintain a supervision during the exercise practice, and a tailoring to obtain more positive results. Future research on physical activity and social network management could be focus also on strength training, to verify if a more structured intervention, consisted of aerobic and strength training, will be able to show an extensive effect.

## Figures and Tables

**Figure 1 ijerph-19-09794-f001:**
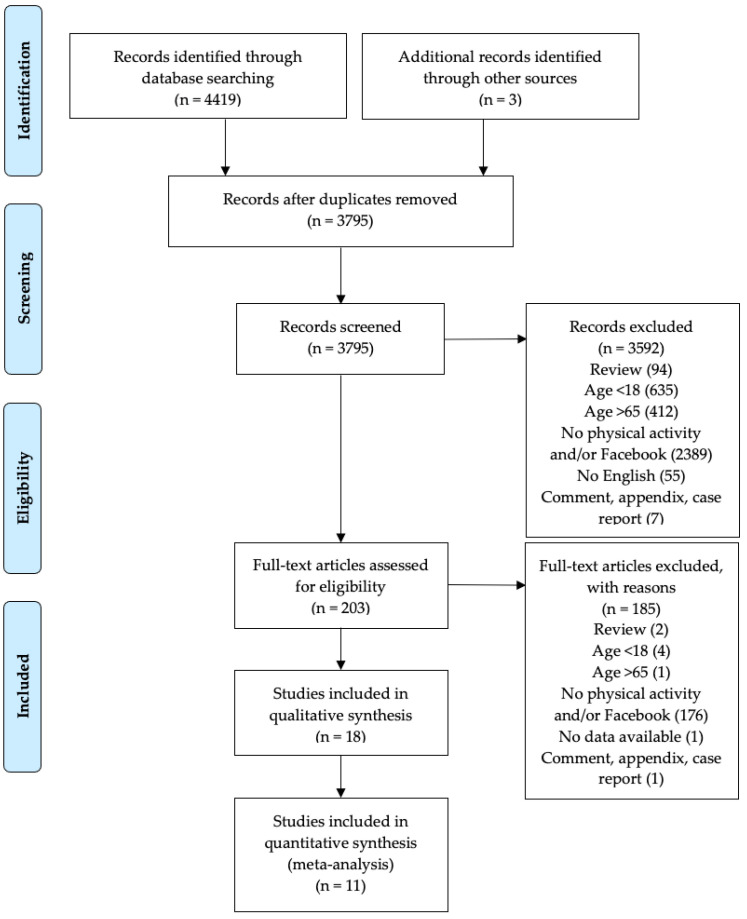
Flow chart of the literature research.

**Figure 2 ijerph-19-09794-f002:**
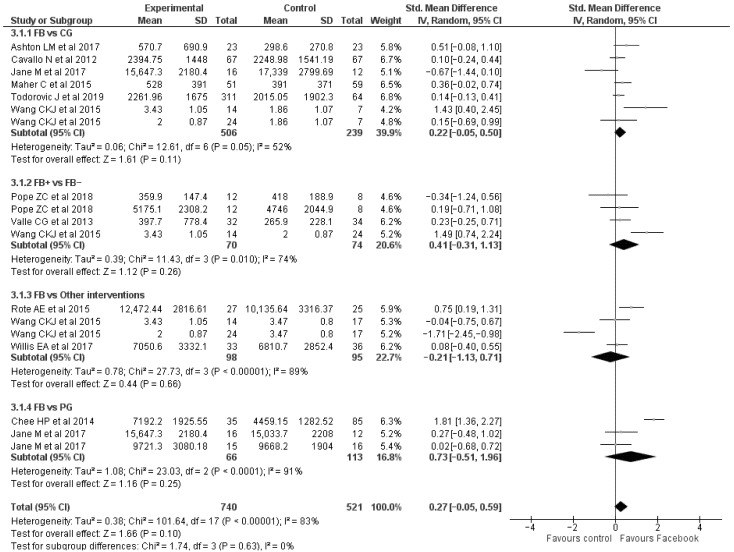
Forest plot representing effect size of FB intervention and control, paper and other groups in total amount of physical activity. FB: Facebook group; CG: control group; PG: paper group; FB+: Facebook with other supports; FB−: Facebook without support.

**Table 1 ijerph-19-09794-t001:** Participants, interventions, comparators, outcomes, study (PICOS) model.

Participants, Interventions, Comparators, Outcomes, Study (PICOS) Designs	
PICOS	Details
Participants	Male and female aged from 18 to 65 years old
Interventions	Supervised or no supervised exercise protocol with Facebook group support
Comparative factors	Facebook-support intervention compared with other type of support
Outcomes	Primary outcomes: evaluation of PA before and after the intervention Secondary outcomes: changes on weight, BMI, body circumferences, body composition, diet parameters, cardiovascular parameters
Study designs	Pilot study, RCT, no-RCT, exploratory study, Randomized pilot trial

PA: physical activity; BMI: Body Mass Index; RCT: randomized controlled trial.

**Table 2 ijerph-19-09794-t002:** Quality assessment of the included studies.

Citation	Randomization Procedure	Similarity of Study Groups	Inclusion or Exclusion Criteria	Dropouts	Blinding	Compliance	Intention-to-Treat Analysis	Timing of Outcomes Assessment	Follow-Up	Results
Todorovic J et al. (2019) [22]	−	+	−	+	−	−	−	+	−	3/9
Pope ZC et al. (2019) [23]	+	+	+	+	−	+	−	+	−	6/9
Torquati L et al. (2018) [24]	−	−	+	+	−	+	+	+	+	6/9
Looyestyn J et al. (2018) [25]	+	−	+	+	−	+	+	+	+	7/9
Pope ZC et al. (2018) [26]	−	−	+	+	−	−	−	+	−	3/9
Pope ZC et al. (2018) [26]	+	+	+	+	−	+	−	+	−	6/9
Naslund JA et al. (2018) [27]	−	−	+	+	−	+	−	−	−	3/9
Krishnamohan S et al. (2017) [28]	−	+	+	+	−	−	−	−	−	3/9
Ashton LM et al. (2017) [29]	+	−	+	+	+	+	+	−	−	6/9
Jane M et al. (2017) [30]	+	−	+	+	−	−	−	+	−	4/9
Willis EA et al. (2016) [31]	+	+	+	+	−	+	+	−	−	6/9
Joseph RP et al. (2015) [32]	+	+	+	+	−	+	−	+	−	6/9
Rote AE et al. (2015) [33]	+	+	+	−	−	+	−	+	−	5/9
Maher C et al. (2015) [18]	+	−	+	+	+	+	+	+	+	8/9
Wang CKJ et al. (2015) [34]	−	−	−	−	−	−	−	−	−	0/9
Chee HP et al. (2014) [35]	+	−	+	+	−	+	−	+	+	6/9
Valle CG et al. (2013) [36]	+	−	+	+	−	+	+	−	−	5/9
Cavallo N et al. (2012) [37]	+	−	+	+	−	−	+	+	−	5/9

**Table 3 ijerph-19-09794-t003:** Characteristics of the inclusion study.

Study	Subjects and Grouping (n)	Protocol Duration and Frequency; Training Modality, Program and Intensity
Todorovic J et al. (2019) [22]	375 medical students FB-G (311) CG (64)	Duration: 4 weeks Facebook group Students were free to practice any kind of physical activity if and when they wanted to. Facebook group was managed by the research team, that used a participatory approach. All members were allowed to post motivational messages or questions for their peers; the reports and photos from organized events were also posted regularly, so all participants could follow the level of participation of other students. Control group In this group, students were free to practice any kind of physical activity if and when they wanted to, but they did not join the facebook group.
Pope ZC et al. (2019) [41]	38 college students, men age 21.5 FB+S-G (19; 15F, 4M) FB-G (19; 13F, 6M))	Duration: 12 weeks Facebook group1 1. Smartwatch to recorded daily step; 2. FB group: Social Cognitive Theory-related health education tips 2 twice-weekly. The aim was to integrate PA into their daily routine improving PA-related self-efficacy, outcome expectancy, social support and enjoyment. Facebook group2 FB group: Social Cognitive Theory-related health education tips 2 twice-weekly. The aim was to integrate PA into their daily routine improving PA-related self-efficacy, outcome expectancy, social support and enjoyment.
Torquati L et al. (2018) [24]	47 healthy nurses, mean age 41.4 FB-G (47; 6M, 41F)	Duration: 12 weeks, with a 6-month follow up. Mobile app to set diet; accelerometers to monitor physical activity behavior; FB group where people share experiences and motivate others. Moreover, diet and PA suggestion was given by FB group.
Looyestyn J et al. (2018) [25]	89 healthy adults, mean age 35.2 FB-G (41; 4M, 37F) PG (48; 29M, 34F)	Duration: 8 weeks, 30 min 3 d/w, with a 5-month follow up. Facebook group Warm-up: 5 min Main part: week 1 (5 sets run/walk 1:1 min; 6 sets run/walk 1:1 min; 5 sets run/walk 1.5:1.5 min); week 2 (6 sets run/walk 1.5:1.5 min; 5 sets run/walk 2:2 min; 3 sets run/walk 3:3 min); week 3 (6 sets run/walk 2:2 min; 6 sets run/walk 2:1 min; 6 sets run/walk 3:1.5 min); week 4 (5 sets run/walk 4:2 min; 7 sets run/walk 3:1 min; 12 min run); week 5 (6 sets run/walk 4:2 min; 10 sets run/walk 3:1 min; 4 sets run/walk 6:2 min); week 6 (15 min run; 3 sets run/walk 8:3 min; 18 min run); week 7 (5 sets run/walk 6:2 min; run 22 min; 3 sets run/walk 10:2 min); week 8 (26 min run; 16 min run, 2 min walk, 16 min run; 30 min run) Cool-down: 5 min Each day the group facilitator posted a message to the closed FB group. These posts were informative and encouraged social interaction including asking participants to post photos, providing information with lonks, motivational quotes, opinion polls, and posts prompting participants to answer questions and interest. Paper group Self-directed running program only to follow individually.
Pope ZC et al. (2018) [26]	8 female breast cancer survivor, mean age 45.8	Duration: 10 weeks: 1. Mobile application to recorded daily step 2. FB group: Social Cognitive Theory-related health education tips 2 twice-weekly. The aim was to integrate PA into their daily routine improving PA-related self-efficacy, outcome expectancy, social support and enjoyment while reducing physical activity-related barriers.
Pope ZC et al. (2018) [26]	20 breast cancer survivor, mean age 52.8 FB+S-G (12F) FB-G (8F)	Duration: 10 weeks Facebook group1 1. Smartwatch to recorded daily step; 2. FB group: Social Cognitive Theory-related health education tips 2 twice-weekly. The aim was to integrate PA into their daily routine improving PA-related self-efficacy, outcome expectancy, social support and enjoyment. Facebook group2 FB group: Social Cognitive Theory-related health education tips 2 twice-weekly. The aim was to integrate PA into their daily routine improving PA-related self-efficacy, outcome expectancy, social support and enjoyment.
Naslund JA et al. (2018) [27]	25 obese people with depressive disorder, bipolar disorder, and schizophrenia, mean age 49.2 FB-G (19; 7M, 12F) CG_not join the Facebook Group (6; 5M, 1F)	Duration: 24 weeks, 1 + 2 d/w Facebook group One weight management session facilitated by two lifestyle coaches and two optional exercise sessions led by a certified fitness trainer. Weight management: interactive focused on healthy eating and exercise, with group discussion and teamwork exercise when participants worked together to plan healthy meals and overcome challenges to adopting healthier lifestyle. Exercise sessions: stretching, resistance and cardio exercise tailored to the needs of obese sedentary adults. These sessions were intended to help participants work towards reaching 150 min of exercise each week. The program also included a secret Facebook group to allow participants to connect and support each other towards achieving their healthy eating and exercise goals.
Krishnamohan S et al. (2017) [28]	45 college students, age: 18–23 FB-G (22; 12M, 10F) CG (23; 12M, 11F)	Duration: 6 weeks Facebook group Inclusion in a private Facebook group in which the health education messages were posted thrice a week in the form of pictures, videos, quotes. Content was decided by a team of dietician, physical education trainer and public health specialist. Control group No such intervention was done for the control group.
Ashton LM et al. (2017) [29]	47 men, age 18–25 FB-G (24M) CG (23M)	Duration: 12 weeks Facebook group 1. Responsive website with relevant information and resources including guidelines, support video and recommended mobile application for improving eating habits, PA, reducing alcohol intake or coping with stress. 2. Wearable PA tracker with associated mobile phone application. 3. 1 h of weekly supervised PA: 40 min of aerobic and strength exercise, 10 min of healthy eating education, 10 min of stress and well-being management 4. An individualized session (week 3) in which personal tailored goals for dietary improvements were set. 5. A private FB group to facilitate social support. 6. Gymstick resistance band for home-based strength training (preferably 2 d/w). 7. A TEMPlate dinner disc to guide main meal portion size form main meal components. Control group Control participants were asked to continue their usual lifestyle for 3 months and offered the program once follow-up assessments were completed.
Jane M et al. (2017) [30]	67 obese or overweight; subjects, mean age: 21–65 FB-G (23; 4M, 19F) PG (23; 2M, 21F) CG (21; 4M, 17F)	Duration: 24 weeks Facebook group Participants were instructed to follow the Total Wellbeing Diet: an energy-reduced, low fat, lower carbohydrate, and higher protein diet. In addition, they were issued with a pedometer and instructed to achieve 10,000 steps per day. The information was received into a secret FB group. Paper group The same instructions and information of the FG, but in written form. Control group Participants were instructed to follow the Australian Government dietary guidelines as well as the National Physical Activity Guidelines for Adults as standard care.
Willis EA et al. (2016) [31]	70 obese people, age: 21–70 FB-G (34; 5M, 29F) PhC-G (36; 6M, 30F)	Duration: 24 weeks Facebook group 24 weekly online modules (1 health lesson post, 1 audio recordings of the phone conference group, 4 comments to highlight the major points of the lesson). Participants were instructed to post a minimum of 4 comments on the message boards per week. Phone conference group 24 meeting (1 evening per week) of group phone conference of 60 min. Participants were encouraged to actively participate and interact with the other.
Joseph RP et al. (2015) [32]	29 healthy women, mean age 35 FB-G (14F) PG (15F)	Duration: 8 weeks Facebook group 1. Weekly PA promotion materials posted on the FB group wall; 2. Discussion topics and participants engagement on the group FB wall: week 1 (overview of the national PA recommendations, health benefits of PA, and PA statistics among African American women); week 2 (developing a PA plan that works for you); week 3 (barriers to PA among African American women and strategies to overcoming barriers); week 4 (developing a social support network to promote PA); week 5 (strategies for incorporating short bouts of PA into your daily routine to achieve national PA recommendations); week 6 (testimonials from African American Women on how they successfully incorporate PA into their daily lives); week 7 (National PA recommendations, barriers to PA among African American Women and strategies to overcoming barriers and incorporate more PA into your life); week 8 (strategies for maintaining a physical active lifestyle after the intervention); 3. Motivational text messages promoting PA: 3 text messages every each week (a. tips on strategies to increase PA throughout the day; b. information on how to overcome barriers to PA; c. reminders of the health benefits of PA; d. motivational/inspirational tips and quotes to participants); 4. Adaptive pedometer-based self-monitoring and goal-setting program: weekly individualized step goals and social reinforcement via email. Printed group 1. Mailed print-based component: booklets mailed every two weeks at home with general information on risk factors for cardiovascular disease, the benefits of PA, tips and strategies to increase daily PA, and encouraged participants to perform a minimum of 150 min of MVPA per week; 2. Static pedometer-based self-monitoring program: participants were instructed to achieve a static goal of 8000–10,000 steps each day.
Rote AE et al. (2015) [33]	53 university students, mean age 18 FB-G (27F) SWI-G (26F)	Duration: 8 weeks Facebook group Each week: personal FB message from the intervention leader requesting a report of their steps/day for the previous week. New step goal each week in according to the previous week steps (10% more steps than the previous mean). Weekly post in the FB page and group with educational information. Standard Walking Intervention group Weekly e-mails with the new step goal in according to the previous week steps (10% more steps than the previous mean), and educational information (the same of FB group).
Maher C et al. (2015) [18]	110 adults, mean age 35.6 FB-G (51; 14M, 37F) CG (59; 18M, 41F)	Duration: 50-day, with a 20-week follow up Facebook group Team challenge: participants are provided with a pedometer and encourage achieving 10,000 steps per day, working in teams of 3 to 8 FB friends. Control group Waiting list, with health monitoring over the ensuing 5 months.
Wang CKJ et al. (2015) [34]	62 university students, mean age 22.3 FB3h-G (14) FB1h-G (24) EG (17) CG (7)	Duration: 8 weeks Facebook1 group 3 h of physical fitness class taught by university lecturer each week + FB group. Facebook2 group Voluntary 1 h of exercise programme taught by experienced university lecturer each week + FB group. Exercise group 3 h of physical fitness class each week. Control group This group did not receive any intervention.
Chee HP et al. (2014) [35]	120 government employees, mean age: 18–59 FB-G (35; 11M, 24F) PG (85, 23M, 62F)	Duration: 16 weeks, with a 2-month follow up Facebook group The participants received a card to log the number of steps taken per day and a pamphlet on PA information that summarized the information provided on FB. Moreover, participants can view and comment the content of the posts, in addition to logging their daily steps counts, through the FB group page. The content of the FB posts were the subsequent: Week 1: healthy every day with 10,000 steps per day Week 2: PA recommendation (10,000 steps per day, or 150 min of moderate intensity PA per week, or 75 min of vigorous PA per week) Week 3–5: benefits of walking Week 6–8: walking as the PA Week 9: PA level based on number of steps per day Week 10–15: strategies to increase number of steps per day (e.g., to park the vehicle far away from the office; to use the stairs instead of the elevator; to take 10-min walks for every 2-h worked) Week 16: PA pyramid Printed group The printed group did not receive a weekly physical activity-related intervention, but the participants obtained a card to log the number of steps taken per day and a pamphlet on PA information.
Valle CG et al. (2013) [36]	86 cancer survivors, mean age 31.7 FB-G.fit (45; 4M, 41F) FB-G.com (41; 4M, 37F)	Duration: 12 weeks Facebook group—fitnet Participants received a pedometer. Each week was posted a Fb message, with expanded behavioral lesson with specific guidance on PA and behavioral strategies, such as enlisting social support, incorporating PA into daily routine, self-monitoring and maintaining PA. The administrator posted various prompts including discussion question, links to videos, exercise- or cancer-related news articles, or electronic PA resources, and a weekly reminder to set an exercise goal, log daily PA. Facebook group—comparison The same intervention but without administration moderation that encourage the interaction.
Cavallo N et al. (2012) [37]	134 university students, aged <25 years FB-G (67F) CG (67F)	Duration: 12 weeks Facebook group INSHAPE website, which provided educational information related to PA and a self-monitoring tool that allowed participants to set goals, track their daily PA, and view a chart depicting their progress relative to their goal and to national recommendations for PA. Control group Control group participants received access to a limited version of the INSHAPE website, which excluded self-monitoring.

d/w: day/week; M: male; F: female; FB-G: Facebook group; PG: paper group; FB+S-G: Facebook + smartwatch group; PhC-G: phone conference group; EG: exercise group; TM-G: text message group; SWI-G: Standard Walking Intervention Group; FB: Facebook; PA: physical activity; MVPA: moderate voluntary physical activity; INSHAPE: Internet Support for Healthy Associations promoting Exercise.

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
