# Peer review of "The Role of Facebook® in Promoting a Physically Active Lifestyle: A Systematic Review and Meta-Analysis"

_ijerph, 2022, doi:10.3390/ijerph19169794_

Round 1
Reviewer 1 Report
The meta-analysis of Duregon et al. investigate the role Facebook in promoting PA. In a society affected by “the physical inactivity pandemic” and where social media play an important role due to the time that people spend on them it is important to better understand the role that they have in promoting PA.
This is an interesting systematic review and the authors have illustrated an innovative topic that adheres well to the journal’s aims. The paper is generally well written and structured.
Abstract:
Clear and well written
Introduction:
- Lines 63-66: please clarify the phrase "physical environments, and policy”. It is suggested to change it into "physical environments and policy" to ensure that policy is part of the fourth general level.
- Line 67: please consider changing the phrase “the mentioned framework suggests” into “it appears to be”, to not mislead readers due to the several references mentioned before.
Materials and method:
- Line 99: please clearly state the literature research final date: a two-month period appears to be unprecise.
- If relevant the authors should provide information on the increased number of articles found in the literature across the years. In 2004 there were the same number of articles as in 2020?
- Line 167: the reference 23 does not contain the author “Sawilowsky”.
- Line 214: the word “in addition” appears to be more appropriate instead of “contrarily”.
Language and grammar
- Please take these minor revisions as an opportunity to thoroughly proofread the manuscript for correct grammar and punctuation:
- Line 70: do you mean “there were considered” instead of “they were considered”. Please, check this.
- Line 78: I would suggest “To the best of the authors’ knowledge” instead of “As the best of authors’ knowledge”.
- Line 90: please check this: “individual” may need the plural form “individuals”.
- Line 123: a “.” after “Cochrane Collaboration Back Review Group [21]” should be added.
Result and discussion:
- The Figure 2 does not clearly explain the “FB-“ at 3.1.2. Not clear what FB+ and FB- have done. Which is the control group? Please include a short description in the legend or in material and methods.
- Tables 3 and 4 were not reported in the reviewed file.
c) Line 245: please clarify what type of “heart rate” the authors meant.
- Lines 288-290: It seems that all the parameters changed at the follow-up by “9.6%, 14.5%, 9.2%, 19.7%, and 11.7%” – not clear if the changes were favorable or not (e.g. cholesterol improved or decreased?).
- Lines 436-437: please clearly state what the authors meant for the phrase “less incisive FB tips”.
- Please discuss in the limitations section the heterogeneity of the methodology to evaluate PA parameters (e.g., pedometers, surveys, field test).
References
The reference style used seems to be non t compliant with the journal guidelines.
Lines 510-511: please double check the “Physical activity strategy for the WHO European Region 2016–2025” pdf link because it seems to not work properly, leading the reader to https://www.who.int/europe/home?v=welcome
Author Response
Reviewer 1
The meta-analysis of Duregon et al. investigate the role Facebook in promoting PA. In a society affected by “the physical inactivity pandemic” and where social media play an important role due to the time that people spend on them it is important to better understand the role that they have in promoting PA.
This is an interesting systematic review and the authors have illustrated an innovative topic that adheres well to the journal’s aims. The paper is generally well written and structured.
Abstract:
Clear and well written
Introduction:
- Lines 63-66: please clarify the phrase "physical environments, and policy”. It is suggested to change it into "physical environments and policy" to ensure that policy is part of the fourth general level.
- Line 67: please consider changing the phrase “the mentioned framework suggests” into “it appears to be”, to not mislead readers due to the several references mentioned before.
Authors: thanks for the suggestions. We change the two sentences.
Materials and method:
- Line 99: please clearly state the literature research final date: a two-month period appears to be unprecise.
Authors: we specify the end of literature research
- If relevant the authors should provide information on the increased number of articles found in the literature across the years. In 2004 there were the same number of articles as in 2020?
Authors: according to authors is not relevant provide information about the increase of article across the years. This increase is nonetheless related to two aspects. The first is that Facebook, although born in 2004, was usable only from computers. The second, is that the potential of Facebook as a system for promoting physical activity can be considered functional with the diffusion of the smartphone. This, in fact, allows continuous feedback wherever the subject is.
- Line 167: the reference 23 does not contain the author “Sawilowsky”.
Authors: we add the author’s name
- Line 214: the word “in addition” appears to be more appropriate instead of “contrarily”.
Authors: we agree and change contrarily with “in addition”.
Language and grammar
- Please take these minor revisions as an opportunity to thoroughly proofread the manuscript for correct grammar and punctuation:
- Line 70: do you mean “there were considered” instead of “they were considered”. Please, check this.
- Line 78: I would suggest “To the best of the authors’ knowledge” instead of “As the best of authors’ knowledge”.
- Line 90: please check this: “individual” may need the plural form “individuals”.
- Line 123: a “.” after “Cochrane Collaboration Back Review Group [21]” should be added.
Authors: thanks for the suggestions. We correct.
Result and discussion:
- The Figure 2 does not clearly explain the “FB-“ at 3.1.2. Not clear what FB+ and FB- have done. Which is the control group? Please include a short description in the legend or in material and methods.
Authors: we specify in the legend the two types of FB group.
- Tables 3 and 4 were not reported in the reviewed file.
Authors: we are sorry for the forgetfulness. We add the two tables in the main text.
- c) Line 245: please clarify what type of “heart rate” the authors meant.
Authors: we specify the type of heart rate
- Lines 288-290: It seems that all the parameters changed at the follow-up by “9.6%, 14.5%, 9.2%, 19.7%, and 11.7%” – not clear if the changes were favorable or not (e.g. cholesterol improved or decreased?).
Authors: we specify how the parameters changed.
- Lines 436-437: please clearly state what the authors meant for the phrase “less incisive FB tips”.
Authors: we better explain this concept.
- Please discuss in the limitations section the heterogeneity of the methodology to evaluate PA parameters (e.g., pedometers, surveys, field test).
Authors: we add heterogeneity of methodologies in the limitation section.
References
The reference style used seems to be non t compliant with the journal guidelines.
Lines 510-511: please double check the “Physical activity strategy for the WHO European Region 2016–2025” pdf link because it seems to not work properly, leading the reader to https://www.who.int/europe/home?v=welcome
Authors: we correct the reference style and verify the link of reference.
Reviewer 2 Report
Line 24 needs to be rewritten or rephrased differently. I believe this may have been a small grammatical error on the part of the authors.
Line 78 I would recommend the authors change "as the best of knowledge" to "Research supports that"
Line 95 is the first place where the authors introduce their model of analysis I would recommend the authors include this in the abstract so the readers can have a clear understanding.
This reviewer recommends that the authors consider offering some potential recommendations or implications for public health praxis based on their research findings. I think it would be insightful to the readers, given how Facebook is a large social media platform and how global influence impacts public health awareness.
Author Response
Reviewer 2
Line 24 needs to be rewritten or rephrased differently. I believe this may have been a small grammatical error on the part of the authors.
Authors: we change the sentence
Line 78 I would recommend the authors change "as the best of knowledge" to "Research supports that"
Authors: we change with “To the best of the authors’ knowledge”, as suggest by reviewer 1
Line 95 is the first place where the authors introduce their model of analysis I would recommend the authors include this in the abstract so the readers can have a clear understanding.
Authors: we add the methods
This reviewer recommends that the authors consider offering some potential recommendations or implications for public health praxis based on their research findings. I think it would be insightful to the readers, given how Facebook is a large social media platform and how global influence impacts public health awareness.
Authors: we implement the conclusion as suggest.